# Interaction of Lipophilic Cytarabine Derivatives with Biomembrane Model at the Air/Water Interface

**DOI:** 10.3390/membranes12100937

**Published:** 2022-09-27

**Authors:** Jhon Fernando Berrio Escobar, Cristiano Giordani, Stefano Russo, Francesco Castelli, Maria Grazia Sarpietro

**Affiliations:** 1Grupo Productos Naturales Marinos, Facultad de Ciencias Farmacéuticas y Alimentarias, Universidad de Antioquia, Calle 70 No. 52-21, Medellín 050010, Colombia; 2Instituto de Física, Universidad de Antioquia, Calle 70 No. 52-21, Medellín 050010, Colombia; 3Dipartimento di Scienze del Farmaco e della Salute, Università degli Studi di Catania, Viale A. Doria 6, 95125 Catania, Italy

**Keywords:** cytarabine, cytarabine derivatives, DMPC, Langmuir–Blodgett, biomembrane models

## Abstract

Cell membrane models are useful for obtaining molecular-level information on the interaction of biologically active molecules whose activity is believed to depend also on their effects on the membrane. Cytarabine was conjugated with fatty acids to improve the drug lipophilicity and the interaction with the biomembrane model. Cytarabine was conjugated with fatty acids of different lengths to form the trimyristoyl cytarabine and the tristearoyl cytarabine derivatives. Their interaction with biomembrane models constituted by dimyristoylphosphatidylcholine (DMPC) monolayers was studied by employing the Langmuir–Blodgett technique. DMPC/cytarabine, DMPC/trimyristoyl cytarabine and DMPC/tristearoyl cytarabine mixed monolayers at increasing molar fractions of the compound were prepared and placed on the subphase. The mean molecular area/surface pressure isotherms were recorded at 37 °C. Between the molecules of DMPC and those of cytarabine or prodrugs, repulsive forces act. However, these forces are very weak between DMPC and cytarabine and stronger between DMPC and the cytarabine derivatives, thus avoiding the expulsion of the compounds at higher surface pressure and modifying the stability of the mixed monolayer. The fatty acid moieties could then modulate the affinity of cytarabine for biomembranes.

## 1. Introduction

Cytarabine, or Ara-C, has been widely used as an anticancer drug since the 1960s for the treatment of many solid and soft tumors in humans [1,2,3,4]. This drug is an analog of cytidine nucleoside and behaves as an antimetabolite agent. Its action mechanisms include inhibition of DNA polymerases and incorporation into the DNA chain, resulting in an incomplete synthesis and demethylation of DNA fragments (epigenetic therapy) and, furthermore, inhibition of cytidine synthesis, which is essential for cell growth. In addition, the nucleoside analogs can cause destabilization of mitochondrial membranes and activation of apoptosis routes [2,3,5].

However, cytarabine has shown several pharmacokinetic disadvantages such as high chemical lability, biological degradation and inactivation in blood plasma; low selectivity in attacking cancer cells; and death of healthy tissue adjacent to the tumor [3,6,7]. Moreover, this drug presents numerous pharmacodynamic problems such as low cellular incorporation across transporter channels, intracellular inactivation and degradation by nucleoside deaminase and ecto-nucleotidase enzymes, respectively, and high elimination through transporter channels and vesicles [3,8,9]. Due to these disadvantages, Ara-C and the other anticancer nucleosides are administrated in high doses and intravenously to keep up the effectiveness, and the treatments are usually very prolonged; these aspects are responsible for many side effects exhibited by this drug, such as gastrointestinal disorder, neurotoxicity, nephrotoxicity and hepatotoxicity, among others. A substantial number of studies are focused on the synthesis of anticancer nucleoside derivatives of cytarabine to improve the chemical, biological and pharmacological properties; in particular, lipophilic conjugates with short- and long-chain fatty acids have been studied [10,11].

The interaction of drugs or prodrugs with the cell membrane is a crucial aspect of explaining and understanding their mechanism of action [12,13,14,15]. However, the study of this interaction is problematic due to the complexity of the eukaryotic membrane structure. Membranes are made of highly dynamic domains, which differ from each other in lipid and/or protein content. Moreover, the lipid fraction is subjected to continuous morphological and behavioral changes since it is composed of phospholipids, organized into a bilayer structure, that deeply modify their disposition and thus their permeability depending on environmental pH, temperature and cholesterol content. Interactions between drug and real biomembranes, firstly due to the protein nature of the latter, are then difficult to study and understand from both biological and physicochemical perspectives [12,13,14,15]. For this reason, it is necessary to employ simplified models of biomembranes made of one or more phospholipids that resemble the main cell membrane’s components; multilayer vesicles (particularly liposomes) and monolayers (e.g., Langmuir monolayers) are widely studied and commonly used as biomembrane models. These simplified models permit easily predicting possible interplays between phospholipids and exogenous molecules with potential biological activity, such as variations induced by these compounds on the diffusion process, their miscibility in bilayers or monolayers, and the overall system stability [12,13,14,15].

Lipid monolayers at the air–water interface can be obtained by Langmuir–Blodgett (LB) and Langmuir–Schaefer (LS) techniques. These monolayers have been used to characterize molecular-level interactions of drugs and have attracted great scientific interest in recent decades, as these architectures offer the advantage of a controllable environment for anchored biopolymers (i.e., proteins and polysaccharides), immobilization of bioactive compounds and manufacturing of hybrid nanomaterials with synthetic polymers or organic compound complexes, therefore offering massive potential for application in several fields [16,17]. The main advantage of the Langmuir monolayers over vesicle or liposomes relies upon their well-defined stable structures: their homogeneity, planar geometry and specific orientation can be employed to study the intermolecular behavior and interactions of biomolecules with the membrane [12,13,14,15]. Langmuir films allow rigorous thermodynamic analysis thanks to a precise modulation of some key parameters such as pH and ion strength. Moreover, lipid monolayers formed at the air–water interface mimic many biological features observed in the natural cells, namely lipid rafts and the interaction of proteins with biological membranes. The main limitation of the Langmuir lipid monolayer as a biomembrane model is its inability to replicate the whole complexity of real biological membranes, rendering the understanding of various membrane functions less accurate [12,13,14,15].

The action mechanism of drugs strongly depends on their interaction with cell membranes. Investigation of such interactions to obtain molecular-level information in living cells is not straightforward, which is why more accessible cell membrane models are employed [12,13,18,19,20,21]. In this study, the interplay between lipophilic derivatives of cytarabine, obtained by conjugation of cytarabine with fatty acids of different lengths, and dimyristoylphosphatidylcholine (DMPC) in mixed monolayers at the air/water interface, was explored. In order to harvest more information and assess the usefulness of the experiments, the interaction of pure cytarabine with DMPC was also investigated. The Langmuir–Blodgett (LB) technique, which measures the surface pressure (i.e., force/length) as a function of the mean molecular area that occupies a molecule in a monolayer extended on an aqueous surface, was extensively applied [19,20,21,22,23,24]. The results showed a bigger effect of the cytarabine derivatives on DMPC monolayers with respect to cytarabine and allowed obtaining precious data on the miscibility/immiscibility of cytarabine derivatives and DMPC in the mixed monolayers. These results could help to better understand the interactions of both cytarabine and cytarabine derivatives with biomembranes, while they could also be used in the design of lipidic carriers for these cytarabine derivatives.

## 2. Materials and Methods

### 2.1. Materials

Cytarabine (purity 99%) (Alfaesar) was used as a substrate; Myristic acid (purity 99%) and stearic acid (purity 99%) were supplied by Merck (Darmstadt, Germany); N,N’-dicyclohexyl-carbodiimide (DCC) (purity 99%) (used as coupling agents for the peracylation reaction) and 4-N,N-dimethylamino-pyridine (DMAP) (purity 99%) (used as a nucleophilic catalyst) were obtained from Alfaesar. The cytarabine derivatives were purified by chromatography and thin-layer chromatography, as previously reported [25]. The cytarabine derivatives were characterized by one- and two-dimensional nuclear magnetic resonance spectroscopy and high-resolution mass spectroscopy in positive mode and direct injection, as reported elsewhere [25]. The 1,2-dimiristoil-sn-glycero-3-phosphatidylcholine (DMPC) (purity 99%) was supplied by Genzyme Pharmaceuticals (Switzerland). Chloroform and tris-hydroximethyl-aminomethane (TRIS) were obtained from Merck (Germany). A film balance apparatus, KSV Langmuir minitrough (KSV, Instruments Ltd., Espoo, Finland), was used. It includes a trough (24,225 mm^2^ available area for the monolayer formation) coated with polytetrafluoroethylene (Teflon) and equipped with a water jacket providing temperature control and two mechanically coupled barriers of hydrophilic Delrin.

### 2.2. Synthesis of Cytarabine Prodrugs

The acylation reaction of the amino group (H_2_N-4) of the cytosine ring and of the hydroxyl groups (HO-4′ and HO-6′) of the arabinose ring of cytarabine with fatty acids was made by Steglich esterification. DCC was added to the solution of fatty acid in a mixture of chloroform/DMF/TEA; the mixture was stirred at room temperature for the activation of carbonyl carbon of organic acid; subsequently, DMAP and cytarabine were added as nucleophilic catalyst and substrate, respectively (see reaction diagram Figure 1). The reaction mixture was stirred at room temperature, as was previously reported [25].

#### 2.2.1. 4-N-Myristoyl-4’,6’-O-dimyristoyl-cytarabine (trimyristoyl–cytarabine)

The product was isolated as a white solid: 1H NMR (CDCl_3_), ẟ ppm (multiplicity; integration; J (Hz); group; position): 0.92 (t; 9H; J = 6.6; -CH_3_; H-14” myristoyl chain), 1.32 (m; 48H; -CH_2_-; H-4” to H13” myristoyl chain, 1.70 (m; 6H; -CH_2_-; H-3” (β protons) myristoyl chain), 2.40 (m; 6H; -CH_2_; H-2” (α protons) myristoyl chain), 4.14 (m; 2H; -CH_2_O-; H-6′ arabinofuranosyl ring), 4.44 (m; 1H; -CH-; H-5′ arabinofuranosyl ring), 4.76 (d; 1H; J = 2.8; -CH-; H-3′ arabinofuranosyl ring), 5.15 (m; 1H; -CH-; H-4′ arabinofuranosyl ring), 6.08 (d, 1H, J = 3.2; -CH-; H-2′ arabinofuranosyl ring), 7.45 (d, 1H, J = 7.8, -CH=; H-5 cytosine ring), 7.90 (d, 1H, J = 7.8, -CH=; H-6 cytosine ring). The 2D NMR characterization and mass spectroscopy values were reported elsewhere [25].

#### 2.2.2. 4-N-Stearoyl-4’,6’-O-distearoyl-cytarabine (tristearoyl–cytarabine)

The product was isolated as a white solid: 1H NMR (CDCl_3_), δ ppm (multiplicity; integration; J (Hz); group; position): 0.87 (t; 9H; J = 6.5; -CH_3_; 18” stearoyl chain), 1.25 (m; 48H; -CH_2_-; 4”-17” stearoyl chain), 1.56 (m; 6H; -CH_2_-; 3” (β protons) stearoyl chain), 2.32 (m; 6H; -CH_2_-; 2” (α protons) stearoyl chain), 4.09 (m; 2H; -CH_2_O-; 6′) 4.35 (m; 1H; -CH; 5′ arabinofuranosyl ring), 4.72 (m; 1H; J = 2.9 -CH-; H-3′ arabinofuranosyl ring), 5.12 (m; 1H; -CH-; H-4′ arabinofuranosyl ring), 6.05 (d, 1H, J = 3.4, -CH-; 2′ arabinofuranosyl ring), 7.49 (d, 1H, J = 7.6, -CH=; H-5 cytosine ring), 7.93 (d, 1H, J = 7.6, -CH=; H-6 cytosine ring). The 2D NMR characterization and mass spectroscopy values were reported elsewhere [25].

### 2.3. Surface Pressure/Mean Molecular Area Isotherms

Film balance measurements were performed using a KSV minitrough apparatus, which includes a trough (24,225 mm^2^ available area for the monolayer formation) coated with polytetrafluoroethylene (Teflon), two mechanically mobile coupled hydrophilic barriers (coated in Delrin), a platinum surface pressure sensor, a computer interface unit and operating software. This system was connected to a circulating water bath to keep the temperature constant at 37 °C (a temperature mimicking the body temperature). The film pressure at the air/water interface was measured using the Wilhelmy plate arrangement attached to a microbalance. A subphase consisting of 5 mM Tris (pH 7.4) solution in ultrapure Millipore water (resistivity 18.2 MΩ cm) was used. The surface purity of the subphase was checked by closing and opening the barriers and ensuring that surface pressure readings were not more than ±0.1 mN/m.

Equimolar solutions of DMPC, cytarabine, trimyristoyl–cytarabine and tristearoyl–cytarabine were prepared in chloroform. Appropriate volumes of DMPC and cytarabine or trimyristoyl–cytarabine or tristearoyl–cytarabine were then mixed to form mixed solutions of DMPC/cytarabine, DMPC/trimyristoyl–cytarabine or DMPC/tristearoil–cytarabine at 0.015, 0.03, 0.045, 0.06, 0.09, 0.12, 0.25 and 0.50 molar fractions of the tested compounds with respect to DMPC. Aliquots of 30 μL of the mixtures, as well as of solutions of the pure components, were spread drop by drop onto the aqueous subphase using a Hamilton syringe. Before use, the Hamilton syringe was cleaned three times with chloroform and then rinsed with the examined solutions. After waiting 15 min for solvent evaporation, the films were compressed with the two mobile barriers at a rate of 10 mm/min, and the surface pressures/mean molecular area isotherms were recorded.

The experiments were performed with constant subphase temperatures of 37 °C (temperature above the phase transition temperature of DMPC) to allow considerations at physiological temperature during which DMPC is in a disordered, permeable state. Each experiment was repeated at least three times to obtain reproducible results.

## 3. Results and Discussion

In this work, solutions of single components (DMPC, cytarabine, trimyristoyl cytarabine and tristearylcytarabine) and solutions of DMPC/cytarabine, DMPC/trimyristoyl cytarabine and DMPC/tristearoyl cytarabine mixtures with different molar fractions of the compounds under examination, were deposited on the subphase to obtain monolayers at the air/water interface. The surface pressure (mN/m)/mean molecular area (Å^2^) curves were recorded at 37 °C, well above the melting transition temperature of DMPC [22]. In fact, at this temperature, the DMPC monolayers behave as a fluid membrane all along the compression isotherm curves. The effect of the compounds on the DMPC monolayer was studied by comparing the DMPC monolayer isotherm with the isotherms of the mixed monolayers.

Figure 2 shows the surface pressure (mN/m)/mean molecular area (Å^2^) isotherms of DMPC, cytarabine and DMPC/cytarabine mixtures (at various molar fractions of cytarabine) monolayers.

The DMPC isotherm has two regions: one between 130 and 115 Å^2^ is characteristic of a gaseous state and the other one, which goes from 115 Å^2^ to lower values of area per molecule, indicates an expanded liquid state. In fact, as the mean molecular area decreases, there is a gradual increase in surface pressure. Cytarabine, up to 20 Å^2^, fails to form a monolayer, as it could disperse in the subphase or remain in the gaseous state. The isotherm curves of the DMPC/AraC mixtures, for low molar fractions of AraC, are very similar to the DMPC isotherm; however, as the AraC mole fraction increases, the curves move towards lower values of the mean molecular area. Starting from 0.25 molar fraction, there is a noticeable displacement of the isotherms towards lower mean molecular area values.

Figure 3 shows the isothermal curves surface pressure (mN/m)/mean molecular area (Å^2^) of DMPC, trimyristoyl cytarabine and DMPC/trimyristoyl cytarabine mixtures at various molar fractions of trimyristoyl cytarabine. Trimyristoyl cytarabine is in an expanded liquid state from 130 to 110 Å^2^ reaching a maximum surface pressure value of about 6 mN/m. This value is maintained for mean molecular area values below 110 Å^2^. These data can suggest that, when the molecules occupy the maximum area, the acylic chains are free to move and affect the surface pressure; as the area decreases and the chains are forced to lift, their freedom decreases, interacting with each other and exerting a constant effect on the surface pressure.

Up to 0.06 molar fraction, trimyristoyl cytarabine does not cause large variations in the isotherms compared to the DMPC one. Starting from the molar fraction of 0.09, trimyristoyl cytarabine causes the shift of the isotherm curves to higher values of the mean molecular area. Furthermore, a transition from the expanded liquid state to the condensed liquid state is evident. The plateau region may also hint at the reorientation of the molecules at the air–water interface or the partial collapse of one of the components of the mixed layer. This transition gradually shifts towards lower surface pressure values. 

Figure 4 shows the surface pressure (mN/m)/mean molecular area (Å^2^) isotherm curve of DMPC, tristearoyl cytarabine and the DMPC/tristearoyl cytarabine mixtures at different molar fractions of tristearoyl cytarabine. Tristearoyl cytarabine, from 130 to about 100 Å^2^, is in an expanded liquid state and reaches a maximum pressure of about 8 mN/m. For mean molecular area values lower than 100 Å^2^, the surface pressure values remain almost constant. At low concentrations of tristearoyl cytarabine, similarly to what we observed for trimyristoyl cytarabine, no major changes were observed in comparison with DMPC isotherm; starting from the molar fraction 0.09 of tristearoyl cytarabine, it is possible to observe a gradual shifting of the isotherms to higher values of area per molecule compared to the pure DMPC isotherm. Initially, the mixed monolayers were in an expanded liquid state; from the molar fraction of 0.06, we observed an expanded liquid/condensed liquid transition, which moves to lower values of surface pressure up to a value of about 15 mN/m for the molar fraction of 0.5. 

In order to obtain more information regarding the type of intermolecular interactions that occur in mixed monolayers consisting of DMPC and the analyzed compounds, the data related to the mean molecular area at different surface pressure values (10 mN/m, 20 mN/m and 30 mN/m) were plotted as a function of the molar fractions of cytarabine, trimyristoyl cytarabine and tristearoyl cytarabine (Figure 5, Figure 6 and Figure 7). In the figures, the dashed lines indicate the values of the mean molecular area at a given surface pressure value of a two-component monolayer with an ideal behavior, thus completely miscible or completely immiscible [26,27,28]. These graphs were obtained by joining the value of the pure DMPC mean molecular area to the one obtained from the pure compound at the surface pressure measure analyzed. Any deviation from the line indicates that the monolayer components are miscible and have a non-ideal behavior.

Figure 5 shows the mean molecular area values of the mixed monolayers DMPC/cytarabine as a function of the molar fraction of cytarabine at 10 mN/m, 20 mN/m and 30 mN/m. It can be observed that cytarabine causes a positive deviation from the ideal values. This behavior, although minimal, is already present in the lower molar fractions and is more evident at the surface pressures of 10 and 20 mN/m. Figure 6 shows the mean molecular area values of the mixed monolayer DMPC/trimyristoyl cytarabine as a function of the molar fraction of cytarabine at the surface pressure of 10, 20 and 30 mN/m. The prodrug causes positive deviations from the ideal line, which become more and more evident for the highest molar fractions at the surface pressure of 10 mN/m. Analogous behavior is observable both at 20 and at 30 mN/m, although at 30 mN/m when 0.25 and 0.5 molar fractions are employed, there is a decrease in the mean molecular area values. Figure 7 shows the trend of the mean molecular area of the DMPC/tristearoyl cytarabine mixed monolayers as a function of the molar fraction of tristearoyl cytarabine at the surface pressure of 10, 20 and 30 mN/m. For low molar fractions (0.015–0.06), there are small positive deviations with respect to the ideal straight line, while for higher molar fractions, positive deviations are very evident. The positive deviation is greater at the surface pressure of 10 mN/m and decreases passing to higher surface pressure values. 

The results obtained indicate that both cytarabine and the two prodrugs interact with the DMPC molecules in the monolayer. Between the molecules of the DMPC and those of cytarabine or prodrugs, repulsive forces of different entities act. In fact, these forces are very weak between DMPC and cytarabine and far stronger in the case of the cytarabine derivatives. In regards to cytarabine, the interaction could be limited to the polar head of the DMPC. The decrease in positive interactions at 30 mN/m could indicate that the high compression removes the cytarabine molecules from the DMPC monolayer, with the consequent dissolution in the sub-phase and the loss of interaction with the DMPC. In the case of the derivatives, on the other hand, the interaction could involve not only the polar head but also the hydrophobic chains of the DMPC; such interactions are present at all surface pressures considered. In previous work, the interfacial behavior of monolayers constituted by DMPC plus gemcitabine or lipophilic gemcitabine prodrugs (4-(N)-valeroyl-gemcitabine, 4-(N)-lauroyl-gemcitabine and 4-(N)-stearoyl-gemcitabine) was studied at the air/water interface. The results showed that the derivatives interact with DMPC producing mixed monolayers that are subject to an expansion effect, depending on the prodrug chain length [29]. However, in that study, the expansion effect was lower than the one caused by trimirystoyl cytarabine and tristearoyl cytarabine, and this behavior could be attributed to the fact that these compounds with three acylic chains have a wider steric hindrance compared to gemcitabine prodrugs with one acylic chain. 

The excess Gibbs free energy of the mixture (Δ*G*, Equation (1)) was calculated by integrating the surface pressure–area (p–A) isotherms of the mixed monolayer and of the pure components, according to the following expression:(1)$$\Delta Gex=\int_{0}^{\pi }\left[A_{12}-\left(X_{1}A_{1}+X_{2}A_{2}\right)\right]\delta \pi$$
where *A*_1_ is the area per molecule of component 1, and *X*_1_ is its mole fraction; *A*_2_ is the area per molecule of component 2, and *X*_2_ is its mole fraction [30]. If the components of the mixed monolayer do not show any specific interactions (i.e., the components form an ideal mixture), the average area per molecule in the mixture is the sum of the areas occupied by each component, and the excess Gibbs energy equals zero. Deviations from ideal behavior give Δ*G* values below or above zero. In particular, negative values of Δ*G_ex_* indicate that the interactions among molecules are more attractive (or less repulsive) as compared to those of their respective standardized pure films, whereas positive values of the Δ*G_ex_* suggest that the interactions in the mixed film are less attractive (or more repulsive) than those of the pure component monolayers. Moreover, positive Δ*G_ex_* values indicate that each component’s molecules interact preferentially with molecules of the same kind. In general, the Δ*G_ex_* values are positive for all the compounds we investigated (Figure 8, Figure 9 and Figure 10). 

This means that the interactions between DMPC and the compounds are more repulsive as compared to the interactions found in their respective pure films. However, differences are evident between cytarabine and cytarabine derivatives, with more repulsive interactions occurring among DMPC and cytarabine derivatives with respect to DMPC and cytarabine. In fact, in the case of cytarabine (Figure 8), we observed that, for the three surface pressures examined, absolute Δ*G_ex_* values varied within an interval of about 400 J/mol, and a maximum point occurs at 0.5 molar fraction with a variation in Δ*G_ex_* of about 125 J/mol between the lowest and highest surface pressures. However, for both trimyristoyl cytarabine (Figure 9) and tristearoyl cytarabine (Figure 10), absolute Δ*G_ex_* values varied similarly within an interval of about 4000 J/mol (which is about 10 times larger than the case of cytarabine) and two maximum points can be observed. The first one is at 0.18–0.25 molar fraction for both 20 and 30 mN/m, while the second maximum point is at 0.5 molar fraction only when the surface pressure is 10 mN/m, and this can be due to a rearrangement of the molecules of the cytarabine derivatives, which favors their packing, and, consequently, there is a lower expansion effect.

The compressibility modulus (Equation (2)), which gives an indication of the elastic/inelastic behavior of the monolayer [31], was evaluated with the following equation: (2)$$C_{s}^{-1}=A\left(\frac{\delta \pi }{\delta A}\right)$$
where *A* is the average area per molecule and *π* is the surface pressure. $C_{s}^{-1}$ depends on the state of the film: the larger the value, the more rigid and less compressible the monolayer is. The reduction in compressibility modulus indicates fluidization of the film [31,32]. The presence of cytarabine in the DMPC monolayer, in general, up to about 20 mN/m, causes a decrease in the compressibility modulus with the result that the mixed monolayers are more compressible and fluid. For higher surface pressure, the compressibility modulus increases with the formation of a more rigid monolayer (Figure 11). Regarding the cytarabine derivatives (Figure 12 and Figure 13), for a lower molar fraction (up to 0.06), they cause an increase in the compressibility modulus, which indicate the formation of a more rigid monolayer. When the molar fraction increases up to certain surface pressure, the compressibility modulus is lower than that of the DMPC; but at high surface pressure, it is higher than that of the DMPC. These data suggest that the compressibility of the mixed monolayer varies as a function of the number of cytarabine derivatives and of the surface pressure. In addition, for a high molar fraction of the compound, two maximum values of the compressibility modulus are displayed, which reflect the phase transition already seen in the isotherms. 

By plotting the excess molecular areas of the mixture as a function of the compounds, the molar fraction can give information on the interactions between components in the mixed monolayers. The excess molecular area was calculated as reported in Equation (3) [33].
(3)$$Aex=A_{12}-\left(X_{1}A_{1}+X_{2}A_{2}\right)$$
where *A*_12_ is the mean molecular area in the mixed monolayers at a given surface pressure; *A*_1_ and *A*_2_ are the molecular area in pure monolayers of 1 and 2 at the same surface pressures, respectively; and *X*_1_ and *X*_2_ are mole fractions of the components in the mixed monolayer. If molecules do no interact or behave as an ideal mixture, the plot of the excess surface area versus monolayer composition yields a straight line; any deviation from linearity indicates that the components are miscible, and the mixed film shows non-ideal behavior. In particular, positive deviations imply repulsive interactions among the film components, while negative deviations imply attractive interactions [33,34]. The positive excess area indicates greater cohesion forces between the same molecules than the different ones, and the negative excess areas indicate greater cohesion forces between the different molecules. For all the three compounds (Figure 14, Figure 15 and Figure 16), positive variations are observed, although in the case of the two cytarabine derivatives, they are about 10 times larger than the case of cytarabine, indicating that repulsive interactions occur between DMPC and the compounds, which can lead to phase separation or partial miscibility. The deviation became smaller as the surface pressure increased. In the case of cytarabine, the presence of two maximum points separated by a minimum point can indicate the formation of domains rich in one or the other component that reveals partial miscibility of the components [35].

## 4. Conclusions

The results obtained in the biophysical evaluations highlight the interactions, behavior and miscibility of cytarabine and its lipophilic prodrugs with DMPC monolayers as models of biomembranes. In this study, the thermodynamic properties of the phospholipid monolayer change considerably with the incorporation of lipophilic derivatives of cytarabine with fatty acids. Langmuir monolayer data suggested a significant molecular-level interaction between acyl derivatives of cytarabine and DMPC; therefore, they are absorbed and dissolved through the biomembrane models [36,37]. The results indicate that the lipophilic derivatives of the nucleoside are able to penetrate into condensed phospholipid monolayers and form stable Langmuir monolayers due to the strong interaction of London and Dipole–Dipole forces among the polar heads and lipophilic chains of amphiphilic prodrugs and DMPC molecules, respectively. The effect exerted by the two cytarabine derivatives is very similar. It can be suggested that the acyl chains of the tristearoyl cytarabine could partly protrude from the monolayer, and the protruding part would not interact with it. In accord with to results shown, the biomembrane models can serve as a simplified cell membrane [36,37]. In this work, two lipophilic derivatives of cytarabine were obtained by conjugation of cytarabine with myristic acid and stearic acid, respectively. The aim was to modulate the affinity with the lipid environment of biological membranes. The study of the interaction of cytarabine and its derivatives with DMPC monolayer at the air/water interface showed that the cytarabine derivatives have a greater effect on the monolayer than plain cytarabine. This is probably due to the acyl chains that allow the derivatives molecules to fit between the lipophilic tails of DMPC molecules.

## Figures and Tables

**Figure 1 membranes-12-00937-f001:**
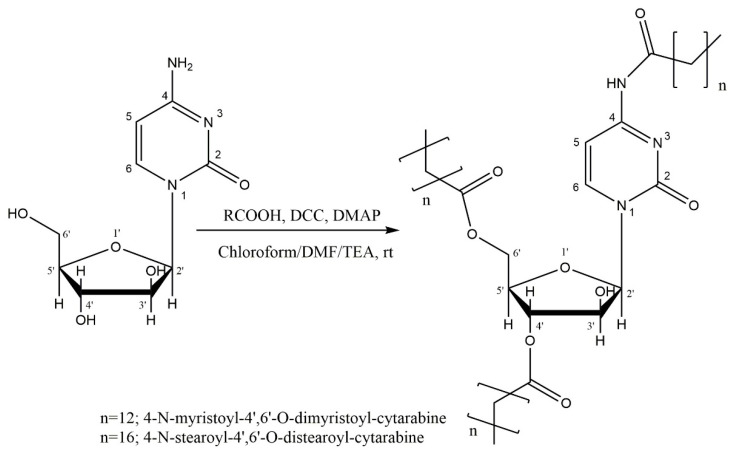
Synthesis of cytarabine derivatives.

**Figure 2 membranes-12-00937-f002:**
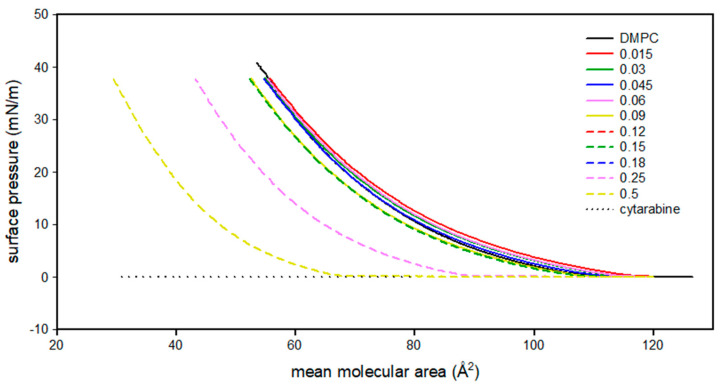
Surface pressure (mN/m)/mean molecular area (Å^2^) isotherm curves of DMPC monolayer in the presence of cytarabine at different molar fractions at 37 °C.

**Figure 3 membranes-12-00937-f003:**
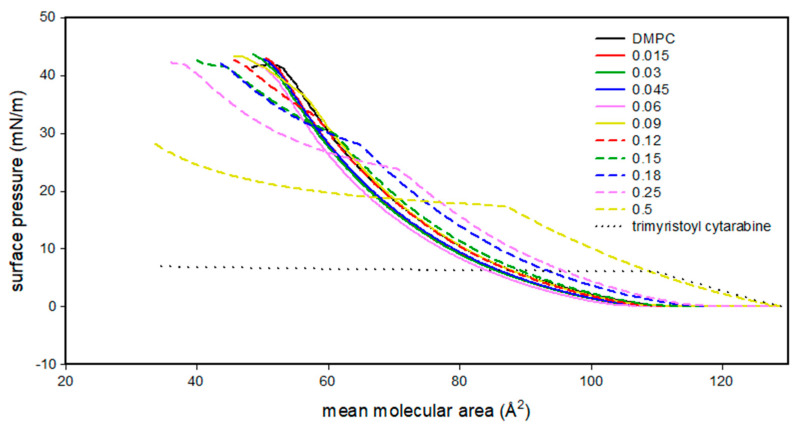
Surface pressure (mN/m)/mean molecular area (Å^2^) isotherm curves of DMPC monolayer in the presence of trimyristoyl cytarabine at different molar fractions at 37 °C.

**Figure 4 membranes-12-00937-f004:**
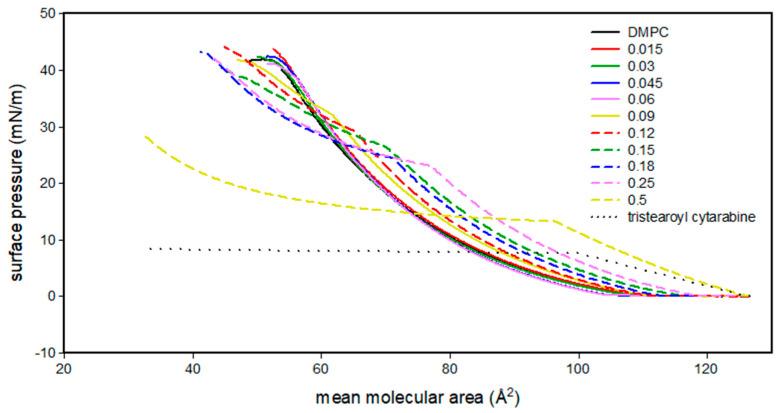
Surface pressure (mN/m)/mean molecular area (Å^2^) isotherm curves of DMPC monolayer in the presence of tristearoyl cytarabine at different molar fractions at 37 °C.

**Figure 5 membranes-12-00937-f005:**
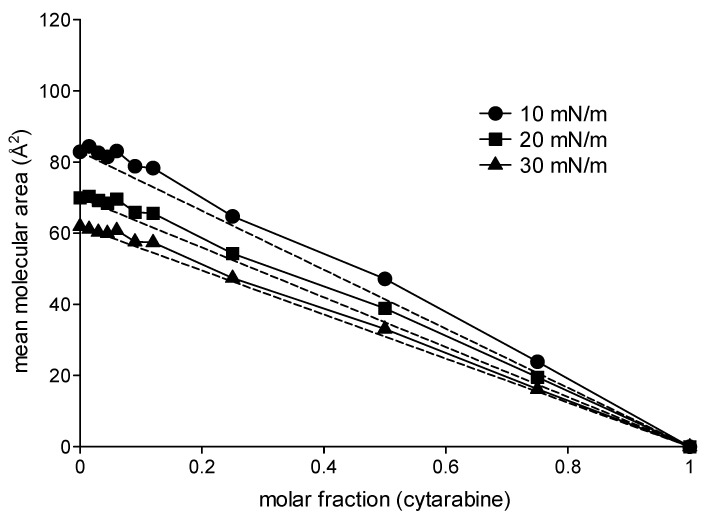
Mean molecular area (Å^2^) of DMPC/cytarabine mixed monolayer as a function of cytarabine molar fractions at 10 mN/m, 20 mN/m and 30 mN/m at 37 °C.

**Figure 6 membranes-12-00937-f006:**
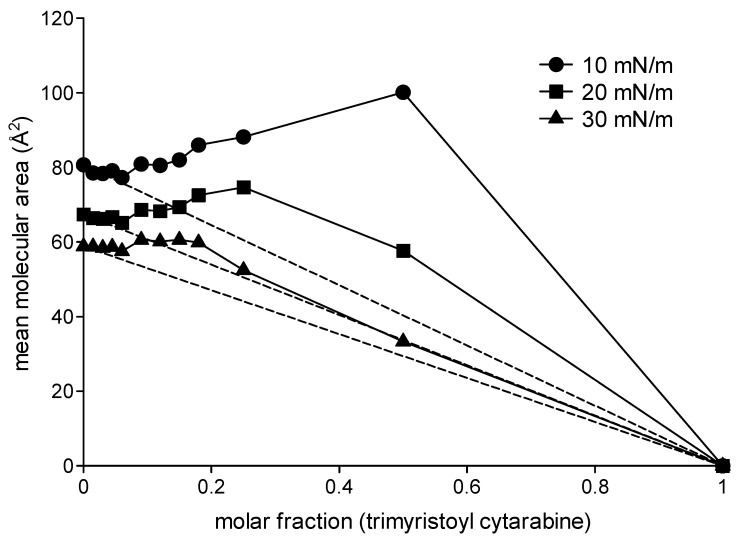
Mean molecular area (Å^2^) of DMPC/trimyristoyl cytarabine mixed monolayer as a function of trimyristoyl cytarabine molar fractions at 10 mN/m, 20 mN/m and 30 mN/m at 37 °C.

**Figure 7 membranes-12-00937-f007:**
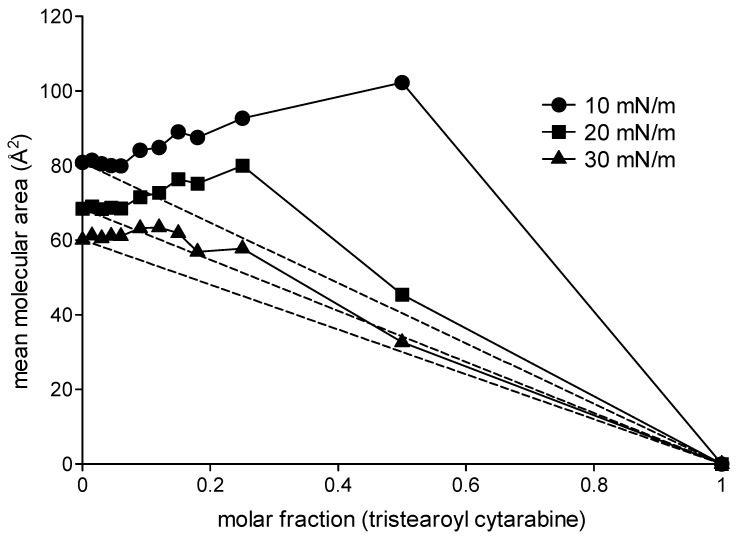
Mean molecular area (Å^2^) of DMPC/tristearoyl cytarabine mixed monolayer as a function of tristearoyl cytarabine molar fractions at 10 mN/m, 20 mN/m and 30 mN/m at 37 °C.

**Figure 8 membranes-12-00937-f008:**
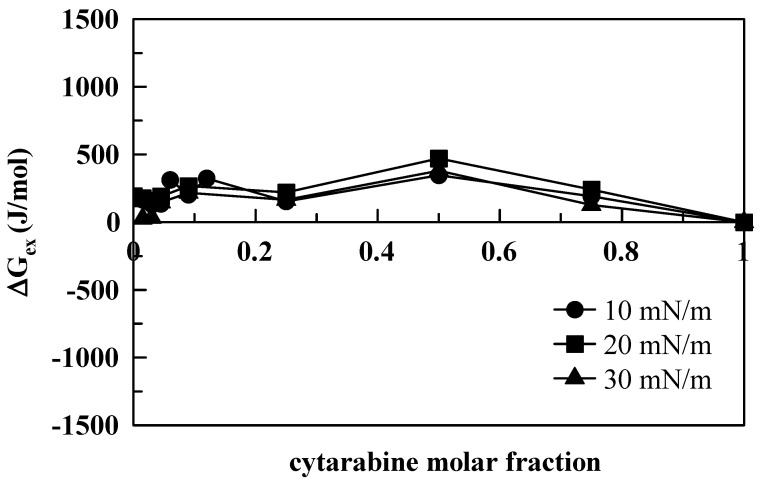
Excess free energy (Δ*G_ex_*) as a function of cytarabine molar fraction, at 10, 20 and 30 mN/m.

**Figure 9 membranes-12-00937-f009:**
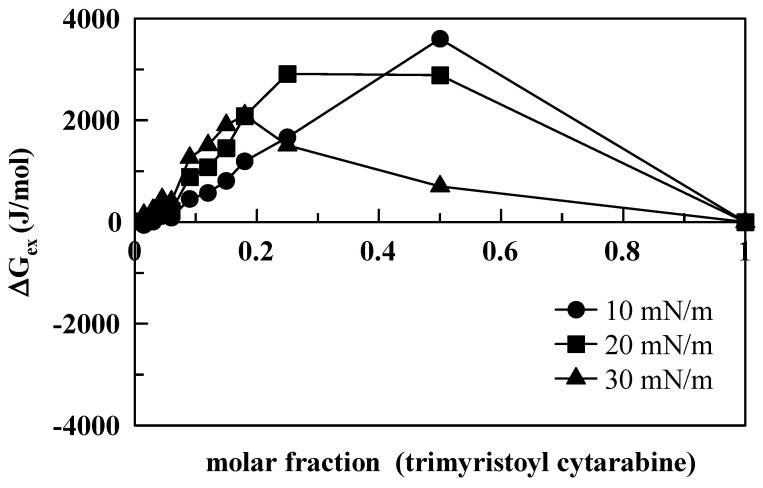
Excess free energy (Δ*G_ex_*) as a function of trimirystoyl cytarabine molar fraction, at 10, 20 and 30 mN/m.

**Figure 10 membranes-12-00937-f010:**
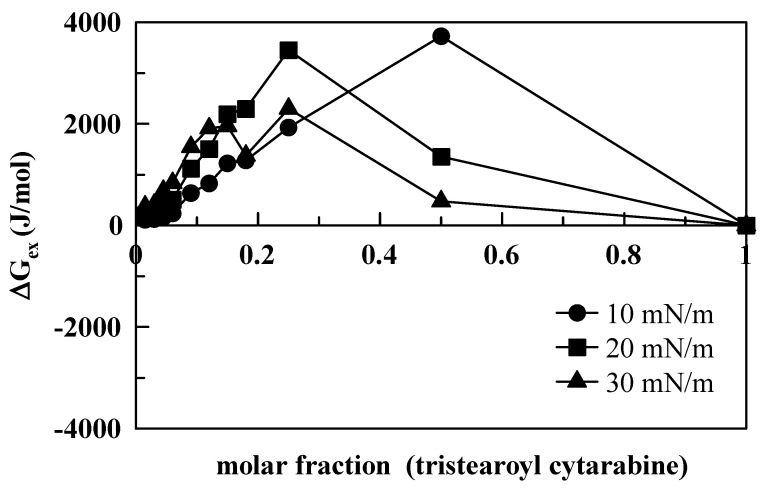
Excess free energy (Δ*G_ex_*) as a function of tristearoyl cytarabine molar fraction, at 10, 20 and 30 mN/m.

**Figure 11 membranes-12-00937-f011:**
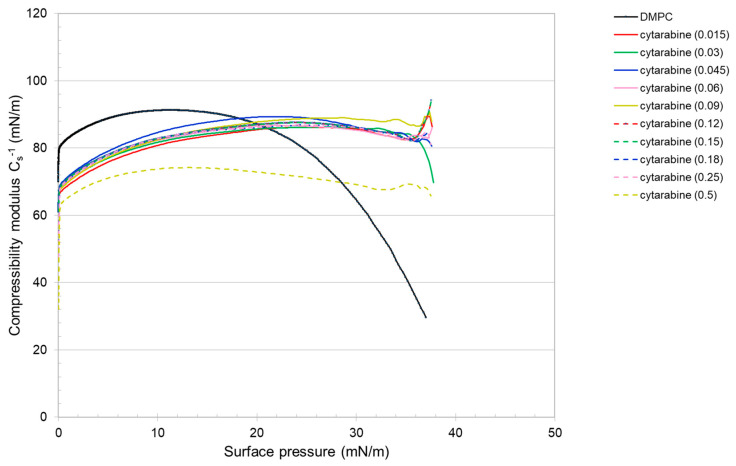
$C_{s}^{-1}$—surface pressure curves of the pure or mixed monolayers made of DMPC and cytarabine.

**Figure 12 membranes-12-00937-f012:**
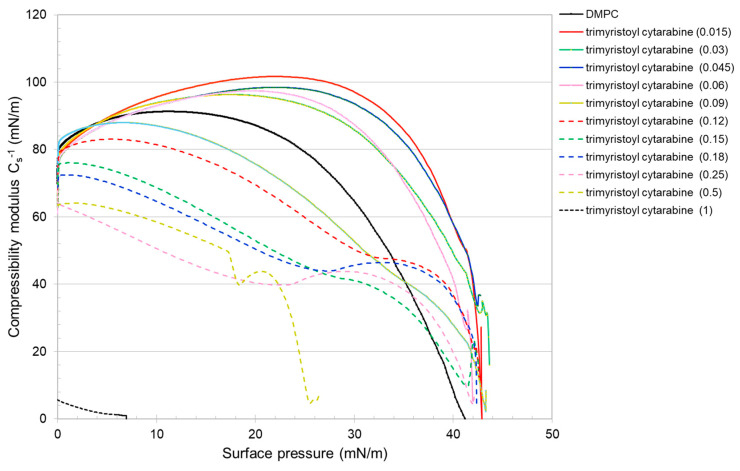
$C_{s}^{-1}$—surface pressure curves of the pure or mixed monolayers made of DMPC and trimyristoyl cytarabine.

**Figure 13 membranes-12-00937-f013:**
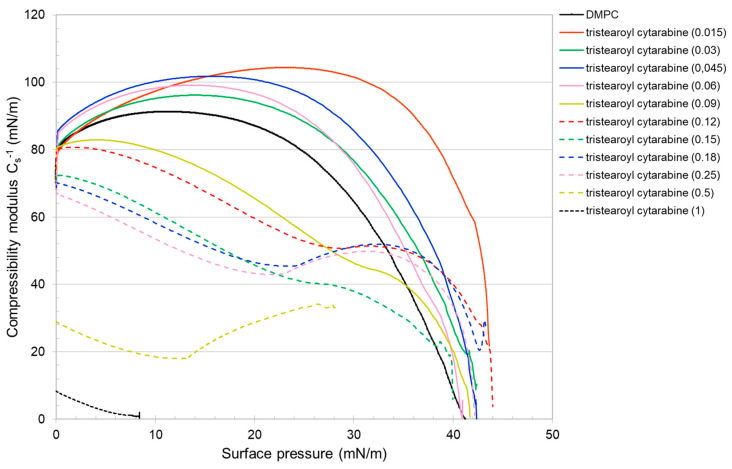
$C_{s}^{-1}$—surface pressure curves of the pure or mixed monolayers made of DMPC and tristearoyl cytarabine.

**Figure 14 membranes-12-00937-f014:**
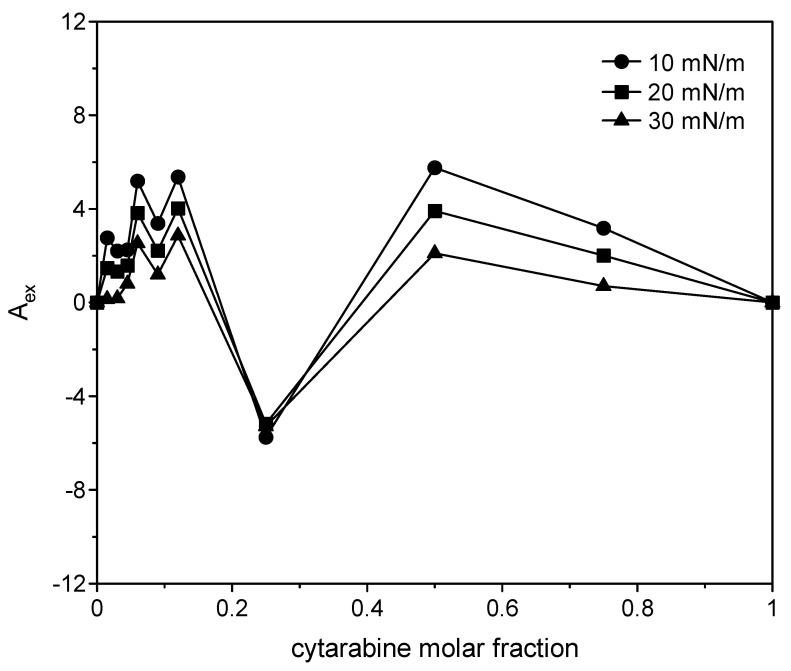
Variation in the excess surface area as a function of cytarabine molar fraction at 10, 20 and 30 mN/m.

**Figure 15 membranes-12-00937-f015:**
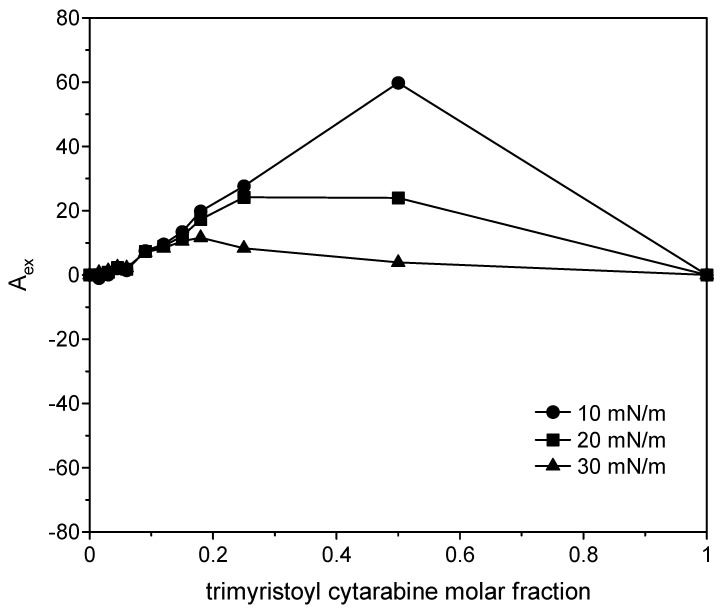
Variation in the excess surface area as a function of trimyristoyl cytarabine molar fraction at 10, 20 and 30 mN/m.

**Figure 16 membranes-12-00937-f016:**
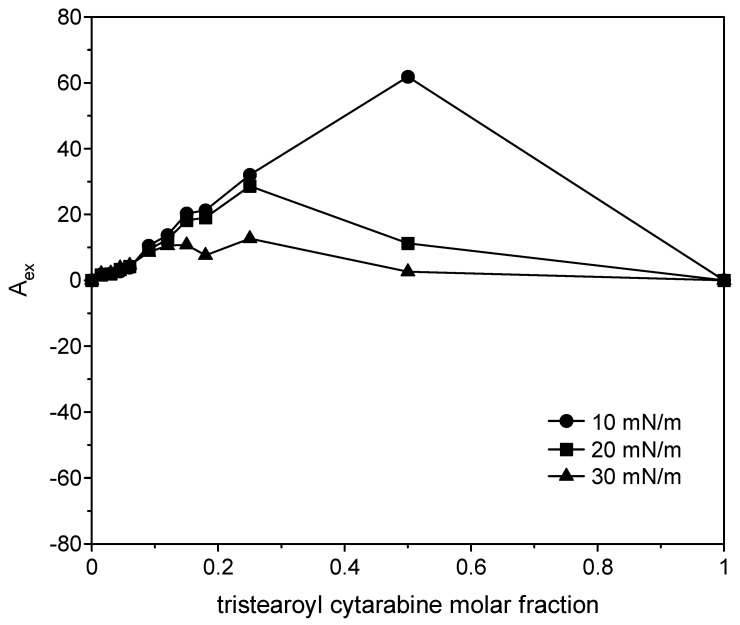
Variation in the excess surface area as a function of tristearoyl cytarabine molar fraction at 10, 20 and 30 mN/m.

## Data Availability

Data were generated at the Department of Drug and Health Sciences, University of Catania. Data supporting the results of this study are available from the corresponding authors on request.

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
