# Peer review of "Interaction of Lipophilic Cytarabine Derivatives with Biomembrane Model at the Air/Water Interface"

_membranes, 2022, doi:10.3390/membranes12100937_

Round 1

Reviewer 1 Report

The authors present an interesting manuscript describing the interaction between lipophilic derivatives of cytarabine, obtained by conjugation of cytarabine with fatty acids of different length and DMPC  at the air/water interface. The rationale for conducting this work is well-justified and the paper aligns with the scopes of the journal.

There are some points that need to be clarified in order to be published in the journal.

1. Please write in a more detailed paragraph why LB technique constitutes an advatangeous method for these biomolecules. It's not very wise to have only a brief description of 2-3 sentences in the introduction (lines 73 and below).

2. Line 145: You should write the exact amount of all aliquots that were used. The phrase of about 30 μl is not appropriate when you use so tiny volumes. The whole process should be reproducable. Please define 

3. Figure 2: Why in figure 2, you do not see the presence of any other phase transitions e.g collapsing of monolayer, solid etc (as you see in figure 3)? This is something that needs to be answered and added in the text. Also in line 171, why cytarabine  fails to form a monolayer or remains in the gaseous state? Please explain.

4. For all isotherms (Fig. 3, 4) you should add the phase transitions. You have only a very general brief description. Please re-write.

5. Did you try to deposit a monolayer on a substrate to see the morphological structures through AFM, SEM etc? It will provide you useful remarks in your systems.

Some references that can help you and you should cite them regarding the phase transitions and the advantages of LB/LS:

doi:10.1021/acsami.2c03944. (Graphene Oxide–Cytochrome c Multilayered Structures for Biocatalytic Applications: Decrypting the Role of Surfactant in Langmuir–Schaefer Layer Deposition).

doi:10.1021/acsomega.9b01970 (Layer-by-layer assembly of clay carbon nanotubes hybrid superstructures).

doi: 10.1016/j.jcis.2018.04.049 (Controlled deposition of fullerene derivatives within a graphene template by means of a modified Langmuir-Schaefer method).

Minor points

Line 74, 208, 212, first line of conclusions: It’s not so appropriate to use “we”. You can use instead passive voice.

The appearance of figures 3,4 is very confusing due to so many isotherms. Please see if you can better adjust the x, y axis accordingly.

Author Response

The English has been extensively revised.

  1. Please write in a more detailed paragraph why LB technique constitutes an advantageous method for these biomolecules. It is not very wise to have only a brief description of 2-3 sentences in the introduction (lines 73 and below).

Answer 1: We have added the advantages of using LB technique in the 3rd paragraph of the introduction.

  1. Line 145: You should write the exact amount of all aliquots that were used. The phrase of about 30 μl is not appropriate when you use so tiny volumes. The whole process should be reproducible. Please define

Answer 2: We have used 30 μl for all the measurements. We have corrected the text in accord.

  1. Figure 2: Why in figure 2, you do not see the presence of any other phase transitions e.g collapsing of monolayer, solid etc. (as you see in figure 3)? This is something that needs to be answered and added in the text. Also in line 171, why cytarabine fails to form a monolayer or remains in the gaseous state? Please explain.

Answer 3: We have added the reason why cytarabine fails to form a monolayer.

  1. For all isotherms (Fig. 3, 4) you should add the phase transitions. You have only a very general brief description. Please re-write.

Answer 4: As requested, we have answered in the manuscript citing two of the three references the reviewer suggested us.

  1. Did you try to deposit a monolayer on a substrate to see the morphological structures through AFM, SEM etc.? It will provide you useful remarks in your systems.

Answer 5: We agree with the reviewer that the analysis of the morphological structure through AFM or SEM could give useful remarks. However, these techniques are not available.

  1. Some references that can help you and you should cite them regarding the phase transitions and the advantages of LB/LS:

Doi:10.1021/acsami.2c03944. (Graphene Oxide–Cytochrome c Multilayered Structures for Biocatalytic Applications: Decrypting the Role of Surfactant in Langmuir–Schaefer Layer Deposition).

Doi:10.1021/acsomega.9b01970 (Layer-by-layer assembly of clay carbon nanotubes hybrid superstructures).

Doi: 10.1016/j.jcis.2018.04.049 (Controlled deposition of fullerene derivatives within a graphene template by means of a modified Langmuir-Schaefer method).

Answer 6: Thank you for your suggestion. We have cited the first and the third references in our manuscript.

Minor points

  1. Line 74, 208, 212, first line of conclusions: It’s not so appropriate to use “we”. You can use instead passive voice.

Answer 7: As suggested, the phrase has been changed.

  1. The appearance of figures 3,4 is very confusing due to so many isotherms. Please see if you can better adjust the x, y axis accordingly.

Answer 8: Figures have been changed. As far as possible, the x-axis has been expanded.

Reviewer 2 Report

The reviewed manuscript investigates interactions of two anticancer drugs, cytarabine derivatives, with model cell membrane formed from DMPC using a Langmuir technique. Although the topic of the studies presented by authors is very interesting and important, the manuscript cannot be published in my opinion. The performed research are incomplete and their analysis is very poor. Below I listed my comments to the manuscript, which maybe allow to improve its quality.

1.      Authors did not use a Langmuir-Blodgett technique in their studies as it is written in line 74. The term Langmuir-Blodgett films is related to the mono- and multilayers deposited onto the solid substrates. The manuscript considers interactions between cytarabine derivatives and DMPC in Langmuir films formed at the air-water interface.

2.      Authors did not provide sufficient state of the art of their research in Introduction. They cite only 16 articles.

3.      Why the authors used in research only one phospholipid? The native cell membrane consists of many phospholipids and other components such as cholesterol, peptides, or sugars. The conclusions about affinity of the investigated cytarabine derivatives cannot be made on the basis of research involving only one phospholipid!

4.      The Langmuir technique gives opportunity to study an adsorption process of bioactive materials to the model cell membrane. Why authors din not take advantage of this opportunity? This kind of an experiment can deliver the most information on affinity of investigated materials to the model cell membrane.

5.      Authors analyze the phase state of the compressed mixed monolayers at the particular compression stages only based on the surface pressure-mean molecular area isotherms. This analysis should be supported by the analysis of the compression modulus-surface pressure dependences, which were not determined by authors.

6.      The conclusions about miscibility of the components in the binary films and interactions between them should base on the analysis of an excess average area per a molecule and excess Gibbs free energy in a function of a molar fraction of  cytarabine derivatives. Moreover, during the compression of the binary films, the Brewster angle microscope images should be recorded in situ to support analysis of the data obtained from isotherms, in particular, when it comes to a monolayer phase state and miscibility of its components.

Author Response

The English has been extensively revised.

  1. Authors did not use a Langmuir-Blodgett technique in their studies as it is written in line 74. The term Langmuir-Blodgett films is related to the mono- and multilayers deposited onto the solid substrates. The manuscript considers interactions between cytarabine derivatives and DMPC in Langmuir films formed at the air-water interface.

Answer 1: In our experiments, we used a Langmuir minitrough to create DMPC lipid monolayers at air-water interface in order to study the interactions between cytarabine derivatives and DMPC as a simple model of lipid membranes.

  1. Authors did not provide sufficient state of the art of their research in Introduction. They cite only 16 articles.

Answer 2: Other articles have been cited.

  1. Why the authors used in research only one phospholipid? The native cell membrane consists of many phospholipids and other components such as cholesterol, peptides, or sugars. The conclusions about affinity of the investigated cytarabine derivatives cannot be made on the basis of research involving only one phospholipid!

Answer 3: We have used only one phospholipid (i.e. DMPC) as a membrane mimetic because DMPC is highly abundant in mammalian membranes and one of the commonly used mammalian membrane mimetics [25-29]. Even though DMPC exhibits some limitations, it is a good candidate for initial studies on interactions between model membrane and proteins.

  1. Langmuir. 2017;33:12351–12361. Doi: 10.1021/acs.langmuir.7b02933.
  2. Chem. Phys. Lipids. 2010;163:480–487. Doi: 10.1016/j.chemphyslip.2010.03.007.
  3. Vib. Spectrosc. 2017; 89:1–8. Doi: 10.1016/j.vibspec.2016.12.006.

28.Biochim. Biophys. Acta (BBA) Biomembr. 2018;1860:654–663. Doi: 10.1016/j.bbamem.2017.12.002.

  1. J. Phys. Chem. B. 2018;122:6236–6250. Doi: 10.1021/acs.jpcb.8b02661.

  1. The Langmuir technique gives opportunity to study an adsorption process of bioactive materials to the model cell membrane. Why authors din not take advantage of this opportunity? This kind of an experiment can deliver the most information on affinity of investigated materials to the model cell membrane.

Answer 4: We agree with the reviewer. Study of the absorption process of some cytarabine derivatives, including those used in this research, is going to be submitted.

  1. Authors analyze the phase state of the compressed mixed monolayers at the particular compression stages only based on the surface pressure-mean molecular area isotherms. This analysis should be supported by the analysis of the compression modulus-surface pressure dependences, which were not determined by authors.

Answer 5: We thank the reviewer for the comment. The analysis of the compression modulus-surface pressure dependence has been added.

  1. The conclusions about miscibility of the components in the binary films and interactions between them should base on the analysis of an excess average area per a molecule and excess Gibbs free energy in a function of a molar fraction of cytarabine derivatives. Moreover, during the compression of the binary films, the Brewster angle microscope images should be recorded in situ to support analysis of the data obtained from isotherms, in particular, when it comes to a monolayer phase state and miscibility of its components.

Answer 6: The analysis of the excess average area per molecule and of the excess Gibbs free energy as a function of the molar fraction of cytarabine and cytarabine derivatives has been added. We agree with reviewer that the BAM images could be useful to support the study. However, the Brewster Angle Microscope is not available. 

Reviewer 3 Report

The issue of the paper is of interest. However there are several questions that the authors should clarify.

The English writing must be extensively revised.

The paper is based only on monolayer experiments, in particular, surface pressure molecular area curves. No analysis of the compresibility are done.

No effects at temperature below Tc.

The effect are visualized as deviations from ideality in the lipophilic derivatives of cytarabine but not by Cytarabine.  No difference if  the lipophilic residue is myritoil or stearic acid. There is no explanation or at least not clear is assayed.

In addition the major effect of both derivatives is around 10mM and decreases at higher concentration. Why is that?

The authors speculate with weak and strong inter4actions but they do not define what are they referring to. In addition, the paper lacks of experimental counter part that may give a molecular insight of these interactions.

Author Response

The English has been extensively revised.

  1. The paper is based only on monolayer experiments, in particular, surface pressure molecular area curves. No analysis of the compressibility are done.

Answer 1: As requested, analysis of the compressibility has been added.

  1. No effects at temperature below Tc.

Answer 2: We did not performed experiments at a temperature below the Tc, as our main purpose was to get information about the interactions between DMPC and compounds at a temperature mimicking the body temperature. This information has been reported in the manuscript.

  1. The effect are visualized as deviations from ideality in the lipophilic derivatives of cytarabine but not by Cytarabine.  No difference if the lipophilic residue is myristoyl or stearic acid. There is no explanation or at least not clear is assayed.

Answer 3: The explanation has been added in the manuscript.

  1. In addition, the major effect of both derivatives is around 10mM and decreases at higher concentration. Why is that?

Answer 4: We understood that the reviewer is talking about 10 mN/m. We answer the question saying that the elevated pressure (> 10 mN/m) causes a rearrangement of the molecules of the cytarabine derivatives, which favors their packing, and, consequently, there is a lower expansion effect.

  1. The authors speculate with weak and strong inter4actions but they do not define what are they referring to. In addition, the paper lacks of experimental counter part that may give a molecular insight of these interactions.

Answer 5: As requested, it has been defined what weak and strong interactions refer to.

Round 2

Reviewer 1 Report

The authors made a significant effort in the revision process. They completed the majority of the comments. For these reasons i recommend publication of this article.

Author Response

Some other improvements have been added in the manuscript.

Reviewer 2 Report

Dear Authors,

I appreciate your efforts and performed extended analysis of the presented research results. However, I still believe that the gathered results are very interesting and deserve deeper and more carefull analysis. I think that also language and style could be improved. 

Best whishes,

Reviewer

Author Response

Dear Reviewer, 

Additional analysis has been added in the results section of the manuscript and English revised.

Reviewer 3 Report

As the author conclude the effects are more significant in the trimyristoil and steroid derivatives than in original cytarabine. This is inferred  from the experimental data reported that the effect is produced by the fatty acid chain rather than by the drug itself.A control with the myristoyl and stearic acids itself is not provided. Could this fatty acids affects the drug interaction? 

I wonder what would be the effect of those additives in the membrane and how it may affect the insertion of the original drug.

The compressibility curves specifically that of DMPC is quite peculiar . Have they checked it with bibliography?

Author Response

QUESTION: As the authors conclude the effects are more significant in the trimyristin and steroid derivatives than in original cytarabine. This is inferred from the experimental data reported that the effect is produced by the fatty acid chain rather than by the drug itself. A control with the myristoyl and stearic acids itself is not provided. Could these fatty acids affect the drug interaction? 

I wonder what would be the effect of those additives in the membrane and how it may affect the insertion of the original drug.

ANSWER: When we use the derivative, the effect observed is not due to cytarabine moiety or fatty acids moieties alone, but to the molecule as a whole.

In this study the control is cytarabine and the aim was to compare the effect of derivatives with that of cytarabine alone;

Of course, the assessment of the effect of the sole fatty acids on DMPC monolayer could be of interest, but the focus of our study was the evaluation of the effect of the conjugation, not of the co-administration.

QUESTION: The compressibility curves, specifically that of DMPC, is quite peculiar. Have they checked it with bibliography?

ANSWER: To our knowledge, there is no other compressibility curve of DMPC in the same condition (37 °C, pH 7.4, Tris 5 mM subphase) available in the bibliography.